# Effect of Al Concentration on Microstructure and Properties of AlNbTiZr Medium-Entropy Alloy Coatings

**DOI:** 10.3390/ma14247661

**Published:** 2021-12-12

**Authors:** Hongyang Xin, Jijun Yang, Jianjun Mao, Qingsong Chen, Jiaqi Yang, Wei Zhang, Zhien Ning, Changqing Teng, Cong Ma, Lu Wu, Xiaoyong Wu

**Affiliations:** 1The First Sub-Institute, Nuclear Power Institute of China, Chengdu 610041, China; xinhongyang2021@163.com (H.X.); maojianjun19840707@163.com (J.M.); 2190016@stu.neu.edu.cn (W.Z.); np201006@163.com (Z.N.); 21626040@zju.edu.cn (C.T.); macong20211113@163.com (C.M.); 2Key Laboratory of Radiation Physics and Technology of Ministry of Education, Institute of Nuclear Science and Technology, Sichuan University, Chengdu 610064, China; jjyang@scu.edu.cn (J.Y.); chqingsong@163.com (Q.C.); yangjiaqi8815827@163.com (J.Y.)

**Keywords:** accident-tolerant fuel, AlNbTiZr MEA coating, corrosion resistance, RF magnetron co-sputtering

## Abstract

The AlNbTiZr medium-entropy alloy (MEA) coatings with different Al contents were prepared on N36 zirconium alloy substrates by RF magnetron co-sputtering. The morphology, microstructure, mechanical properties, surface wettability and corrosion resistance of the AlNbTiZr MEA coatings were studied to evaluate the surface protection behavior of zirconium alloy cladding under operation conditions of a pressurized water reactor. The results showed that all the coatings were composite structures with amorphous and bcc-structured nanocrystals. With the increase of Al content, both the elastic modulus and hardness decreased first and then increased. The hydrophobicity of the coatings was enhanced compared with that of the substrate. The 10.2 at.% Al AlNbTiZr coating had the best corrosion resistance and the minimum oxygen penetration depth, which originated from the formation of a denser oxide layer consisting of Nb_2_Zr_6_O_17_ and ZrO_2_. This study provides an improved idea for the design and development of Al-containing MEA coating materials for accident tolerant fuel.

## 1. Introduction

After the Fukushima nuclear accident, scientists have carried out extensive research on accident tolerant fuel (ATF) projects for pressurized water reactors (PWRs) [1]. Under the loss of coolant accident (LOCA), the cladding materials will be oxidized rapidly and cause immeasurable loss [2]. In comparison, ATF cladding can delay the time of fuel rod swelling explosion and reduce the rate of cladding temperature rise, which can greatly delay the progress of severe accidents [3]. For ATF cladding, the most critical demand is to reduce the high-temperature steam oxidation rate when an accident happens [4]. The main research direction of ATF cladding is divided into two strategies: the first one is to develop new cladding materials instead of zirconium alloys, such as FeCrAl alloy, SiC_f_/SiC non-metallic composites materials and Mo alloys [5,6,7,8,9,10]. The second one is to prepare protective coatings on the surface of zirconium alloy cladding to ameliorate its high-temperature oxidation resistance, mainly representing metallic coatings (Cr, Mo, CrAl, FeCrAl, high entropy alloy, etc.) [11,12,13,14], ceramic coatings (CrC, SiC, CrN, MAX phase materials, etc.) [15,16,17,18,19,20] and composite coatings (Cr/CrN, TiN/TiAlN, etc.) [20,21]. Based on these results, although the above coatings exhibit excellent corrosion resistance in high-temperature pressurized water experiments, it is still an important topic to exploit new candidate coatings that can withstand LOCA accidents in PWR.

In recent years, medium-entropy alloys (MEAs) have attracted increasing research attention. It is usually composed of 2~4 main elements and its mixing entropy (*ΔS_mix_*) in the range of 0.69 *R*~1.61 *R* (*R* is the gas constant) [22,23,24,25]. As one kind of multi-principal element alloy with relatively few main elements and easily controlled chemical composition, it has ultra-high strength, high hardness, excellent corrosion resistance and super irradiation resistance, which is expected to be industrialized and applied in the field of nuclear materials [26,27,28]. Similarly, the MEA coatings also show similar advantages to their bulk materials counterparts. For instance, Feng et al. found that the corrosion resistance of CrCoNi MEA coating prepared by laser cladding was even higher than that of 304 stainless steel counterparts in 0.5 M H_2_SO_4_ solution [29]. Chai et al. found the CrNiTi MEA coating exhibited superior wear resistance and reported an approximately 10-fold improvement in wear resistance over the steel substrate [30]. Cao et al. [27] fabricated a multilayered CoCrNi/Ti MEA coating with nano-twin boundaries, observed in columnar CoCrNi grains, which helped to redistribute the stress and change plasticity of the coating. Wang et al. reported that the NiCoCr MEA exhibited excellent irradiation resistance compared to Ni under the same irradiation conditions, stemming from the random nature of the lattice position of MEA, which suppressed radiation-induced damage [28]. These properties could make the MEA coatings promising as a candidate material for surface protection of PWR structural materials.

In our work, the novel AlNbTiZr MEA coatings with various Al content were designed and investigated for the ATF cladding coating. Considering the application requirements of PWR under accident conditions, four metal elements (Al, Nb, Ti and Zr) were deliberately selected for the following causes [31,32,33,34,35]: (a) Avoid choosing elements that can undergo nuclear transmutation reactions with neutrons, and these selected elements should have low neutron absorption cross-sections, i.e., high melting point elements: Al (0.22 b, 660 °C), Nb (1.1 b, 2415 °C), Ti (5.6 b, 1670 °C) and Zr (0.18 b, 1850 °C). (b) Al, Nb, Ti are strong oxide-forming elements, which can form the dense oxide layer with excellent corrosion resistance. (c) The Al and Nb element show a strong tendency to form a bcc crystal structure. The bcc structure coating materials have good resistance to radiation expansion and radiation-induced segregation. (d) The choice of Zr element is to improve the coating-substrate bonding force by enhancing the metallurgical bonding. What calls for special attention is that the effect of Al content needs to be further explored because of its two-folds on the microstructure and properties of the coating system. On the one hand, as a strong oxidation-forming element, Al can easily form a dense oxide layer on the coating surface, which is beneficial for enhancing corrosion resistance. On the other hand, in critical water and supercritical water corrosion conditions, the Al_2_O_3_ easily reacts with water to generate the boehmite phase (AlOOH) and leads to the dissolution of Al_2_O_3_, which will greatly reduce the corrosion resistance [23,24]. Hence, the effect of Al content on the microstructure, wettability, mechanical properties and corrosion resistance of the AlNbTiZr MEA coatings were systematically evaluated.

## 2. Experimental Details

### 2.1. Coating Deposition

The five AlNbTiZr MEA coatings with different Al content were deposited on single-crystalline silicon wafers and N36 Zr alloy (Zr-1Sn-1Nb-0.3Fe in wt.%) substrates by RF magnetron co-sputtering. The size of zirconium alloy substrates was 10 × 10 × 0.5 mm^3^. The MEA targets with various Al contents were prepared by powder metallurgy technology using alloy with a purity of above 99.99%. All targets were 76.2 mm in diameter and 5 mm in thickness. The content of the Al element was set to 0 at.%, 10 at.%, 15 at.%, 25 at.% and 50 at.%, respectively. The other three elements were equal molar ratios. Before each deposition, the substrates were polished by using different grit SiC abrasive papers (grit size up to 3000#) and then ultrasonic cleaned with acetone to remove surface impurities. During the deposit process, the sputtering power and bias voltage of the AlNbTiZr MEA targets were fixed at ~200 W and −100 V, respectively. The target-substrate distance was kept at 9 cm. The base pressure of the deposition chamber was lower than 7.5×10^−4^ Pa, and the sputter pressure was maintained at 0.51 Pa. The Ar flow rate and the deposition time were controlled at 51 sccm and 150 min, respectively. The single-crystalline silicon and zirconium alloy substrates were rotated at 20 rpm to ensure the uniform thickness of all coatings. Deposition temperature was performed at 300 ℃ in the chamber. The detailed coatings preparation parameters are presented in Table 1.

### 2.2. Coating Characterization

The X-ray diffractometer (XRD, EMPYREAN, PANalytical, The Netherlands) was performed to characterize the crystalline phase of the as-deposited and corroded coatings. The scan range was from 20° to 90° with a resolution of 0.02° per step and a time step of 0.5 s. The microstructure of the coatings was characterized by transmission electron microscopy (TEM, Talos F200S, FEI, Hillsboro, OR, USA). The cross-sectional TEM coating samples were prepared by focused ion beam (FIB, Helios NanoLab 600i, FEI, Hillsboro, OR, USA) electron microscopy. The surface and cross-sectional morphologies, chemical composition and thickness of all the coatings were observed and calculated by field-emission scanning electron microscopy (FESEM, JSM-7500F, JEOL, Japan) equipped with an energy dispersive spectrometer (EDS, Ultim Extreme, Oxford, UK). A nanoindentation instrument (NHT2, Anton Paar, Graz, Austria) with the continuous stiffness method was applied to evaluate the hardness and elastic modulus of the coatings. Each sample was conducted five times, and then its average value was used. The indentation depth should be no more than 1/7 of the coating thickness to eliminate and reduce the substrate effect. The liquid-solid contact angle tester (JC2000D, POWEREACH, Shanghai, China) was used to characterize the surface wettability of the coatings. The corrosion experiments were operated in deionized water, concluding 1000 ppm boron, 3.5 ppm lithium at 360 °C for 30 days. The saturation pressure of the static autoclave was 18.6 MPa at this corresponding temperature value. The oxides scale thickness of the internal coatings was tested to evaluate the corrosion resistance of AlNbTiZr MEA coatings with different Al contents.

## 3. Results and Discussion

### 3.1. Chemical Composition, Phase and Microstructure

To verify whether the composition of coatings was consistent with the expected, the chemical compositions of all the coatings were characterized by EDS. The chemical composition (at.%), thickness and corresponding mixing entropy (∆*S_mix_*) of coatings are listed in Table 2. The contents of the Nb, Ti and Zr in AlNbTiZr MEA coatings remained approximately equal in molar ratios. The cross-sectional SEM images showed that the thickness of all the coatings was between 5.4 μm and 6.1 μm. The slight deviation of coating thickness and composition could result from the different sputtering rates of each element under the influence of Ar^+^ bombardment. As shown in Table 2, the Al content in AlNbTiZr MEA coatings were 0 at.%, 10.2 at.%, 15.6 at.%, 26.2 at.% and 54.7 at.%, respectively. The above results demonstrated that the as-prepared coatings meet the expectation. For the sake of simplicity, one equimolar NbTiZr MEA coating without Al element was named as Al0%. With the increase of Al content, four AlNbTiZr MEA coatings were named as Al10%, Al15%, Al26% and Al54%, respectively. The mixing entropy values (Δ*S_mix_*) were calculated, which increased with the increase Al content from 1.10 *R* to 1.39 *R* and then decreased to 1.19 *R*. Here, the gas constant, *R*, is equal to 8.314 J∙K^−1^∙mol^−1^. It shows that the addition of Al into the NbTiZr coating system still follows the rule of medium-entropy alloys (0.69 *R* ≤ ∆*S_mix_* ≤ 1.61 *R*). Figure 1 shows the surface and cross-sectional SEM morphologies of the as-deposited AlNbTiZr MEA coatings with various Al content. These images exhibited that the coatings were homogeneous and continuous, and no obvious defects, columnar crystals or holes were observed. Small circular particles appeared on the surface of all coatings. With the further addition of Al content, the SEM micrograph at high magnification revealed that the surface of the coatings became smoother, as presented in Figure 2. The results indicated that the coatings have a high deposition quality, which can be beneficial to improve the high-temperature pressurized water corrosion resistance.

Figure 3 exhibits the XRD patterns of the as-deposited AlNbTiZr MEA coatings with different Al contents. The as-deposited AlNbTiZr MEA coatings had distinct wide diffraction peaks between 36.1° and 38.2°, which implies a low crystallinity [36,37]. According to earlier studies, the broad halo diffraction peaks of these MEA coatings were usually caused by the formation of an amorphous or nanocrystalline phase [38,39]. With the increase of Al content in the coating system, the center position of broad diffraction peaks shifted to a higher angle, indicating that the average interatomic distance became smaller, i.e., a compression occurred. This could be due to the increase of lattice distortion caused by the addition of Al atoms with a smaller atomic radius. To further verify and determine the phase structure of the coatings, TEM analyses were performed on all the as-spun coatings, as shown in Figure 4.

The cross-sectional HRTEM images showed that all the coatings were mainly amorphous structures wrapped with tiny nanocrystals. The increase of atomic size difference in the coatings led to the collapse of distorted lattices and promoted the formation of an amorphous phase, which was one of the reasons that the microstructure of all the coatings was dominated by the amorphous phase. From Figure 4(a1)–(e1), the selected area electron diffraction (SAED) pattern exhibited three diffraction rings, corresponding to a stronger diffraction ring of the (100) plane and the weaker diffraction rings of the (200) and (211) planes of the bcc nanocrystalline structure were determined. Meanwhile, the number of bcc nanocrystals in the coatings increased significantly with the increase of Al content. This is not surprising because the Al element shows a strong tendency for forming a bcc crystalline structure. The microstructure of the AlNbTiZr MEA coatings may be due to the inhibition of grain growth by the non-thermodynamic equilibrium magnetron sputtering technology and the combined effect of entropy enhancement in the coatings [40,41]. Therefore, it can be seen that the results of the TEM analyses were in line with those of the XRD analyses.

### 3.2. Mechanical Properties

The mechanical properties of the AlNbTiZr MEA coatings with different Al contents were evaluated by nanoindentation tests. The value of hardness and elasticity modulus as a function of the Al content in the MEA coatings are presented in Figure 5a. To clarify the ability of the coatings to resist cracking and plastic deformation, the *H*^3^*/E**^2^ ratio and *H/E* ratio were calculated [*H* is the nano-hardness, *E* = E/(1 − v^2^*) was the effective elasticity modulus with Poisson’s rate, *v* = 0.3], and the results are shown in Figure 5b. From Figure 5a, the hardness and elastic modulus decreased slightly initially and then gradually increased with the increasing of the Al content in the coating system. The Al content had little effect on the elastic modulus and hardness of the Al0%, Al10% and Al15% coatings. When the elastic modulus of different materials was similar, the materials with lower hardness tended to have lower brittleness and higher fracture toughness. This is because the stress wave propagation distance of the materials with lower hardness is short, and the local energy dissipation is large when acted on by external forces [42]. With the further increase of Al content in the coating, the value of hardness and elastic modulus increased successively. This may be related to the number of bcc nanocrystals in the coatings. The coatings containing more nanocrystals would have more grain boundaries, and the interaction with fine grain strengthening and solid solution strengthening would improve the hardness and strength of the coatings. Compared to other coatings, the Al10% coating was less likely to cause overall failure due to its relatively low hardness and modulus of elasticity. Figure 5b shows the values of *H/E* and *H*^3^*/E**^2^ ratios of all the as-prepared coatings. As is well-known, the value of *H/E* and *H*^3^*/E**^2^ ratios are the key parameters to evaluate the cracking and plastic deformation resistance of coatings, which are also closely related to the tribological properties of coatings. The high *H/E* and *H*^3^*/E**^2^ ratio values mean that the material has stronger resistance to deformation. As shown in Figure 5b, the value of *H/E* and *H*^3^*/E**^2^ ratios increased with the increase of Al content in the coatings, indicating that the anti-deformation ability was enhanced, but the overall difference of their values was not significant.

### 3.3. Surface Wettability

To assess the surface wettability of the AlNbTiZr MEA coatings with different aluminum contents, two microliters of deionized water were dropped on five as-spun coatings and N36 substrate, and their liquid-solid contact angles were tested. In order to reduce the experimental error, each sample was tested five times, and the average was taken. As shown in Figure 6, the contact angle of the substrate was 72.2° ± 4°, indicating a hydrophilic surface. For Al 0%, Al 10%, Al 15%, Al 26% and Al 54% coatings, the contact angles were 84.2° ± 3°, 88.8° ± 2°, 88.5° ± 4°, 88.5° ± 3° and 87.7° ± 3°, respectively. The contact angles of AlNbTiZr MEA coatings with different Al contents were approximately close, indicating that Al content had little influence on the contact angle of the coatings. Further, compared with the contact angle of the N36 substrate, the Al10% coating exhibited a stronger hydrophobicity due to its relatively large contact angle, which may effectively improve the corrosion resistance of the coating [43].

### 3.4. Autoclave Test

To clarify the high-temperature pressurized water corrosion resistance of the AlNbTiZr MEA coatings with various Al contents, the uncoated N36 substrate was also exposed to the same corrosion conditions for comparison. The surface morphologies of the corroded N36 substrate and corroded MEA coatings with different Al content are shown in Figure 7. Figure 7g–l shows an enlarged view of Figure 7a–f. After the corrosion of the N36 substrate, the disc-shaped oxide particles with different sizes were formed on the surface, as shown in Figure 7a. The surface morphologies of the AlNbTiZr MEA coatings with different Al content were obviously different after corrosion. Figure 7b,d shows that the dispersed oxide products were formed on the surface of Al 0% and Al 15% coatings. From Figure 7c,e,f, the uniform oxide scale was formed on the surface of Al10%, Al 26% and Al 54% coatings, which were square-shaped oxide particles on the Al 10% coating, although the oxide particles on the Al 26% and Al 54% coatings were mostly round. The oxide layers of different densities were formed on the surface of the coatings without cracks, delamination or peeling, which indicated that all the coatings have excellent adhesion and durability under high-temperature pressurized water corrosion environments. According to Figure 7, the chemical composition of the surface of each coating was also indicated. It can be found that the atomic percentage of oxygen in all the coatings was not very different, at about 65 at.%, to a certain extent, indicating that the oxygen content is basically saturated.

The phase structure of the corroded N36 substrate and corroded coatings were analyzed by XRD to ascertain the corrosion products, as indicated in Figure 8. The corrosion products of the substrate and AlNbTiZr coatings with various Al contents were ascertained to be orthorhombic Nb_2_Zr_6_O_17_ (PDF card index: 09-0251), cubic ZrO_2_ (PDF card index: 49-1642), which is consistent with the results of corrosion of AlCrMoNbZr HEA coating studied by Zhang et al. [23]. The order of the Gibbs free energy of formation of the involved oxides was Nb_2_O_5_ < Al_2_O_3_ < ZrO_2_ < TiO_2_ [44]. The formation order of corrosion products of the AlNbTiZr MEA coatings should be consistent with the order of formation of Gibbs free energy. The diffraction peaks of ZrO_2_ were observed in all coatings after corrosion, and it can be assumed that Nb_2_O_5_ and Al_2_O_3_ have been formed. Nevertheless, no diffraction peaks were observed for Nb_2_O_5_ and Al_2_O_3_. There are two main reasons for the absence of Al_2_O_3_ diffraction peaks in the XRD patterns of Al-containing MEA coatings: (1) Al_2_O_3_ has a higher crystallization temperature, the formation temperature of γ-Al_2_O_3_ is 450° and higher for α-Al_2_O_3_ [45]. (2) Al reacts easily with water to form the boehmite phase (AlOOH), which is partially dissolved in the high-temperature aqueous corrosion environment [31]. Meanwhile, Nb_2_O_5_ can dissolve with ZrO_2_ to form the Nb_2_Zr_6_O_17_ phase, which confirmed the test results in XRD patterns. The Nb_2_Zr_6_O_17_ phase has a high melting point, which can be used as a barrier to prevent the diffusion of oxygen into the N36 substrate, which can be effectively applied in LOCA conditions [46,47]. It is worth noting that the characteristic broad halo diffraction peaks of the AlNbTiZr MEA coatings were still identifiable after corrosion, indicating the coatings exhibited excellent structural stability.

For further determining the high-temperature pressurized water corrosion resistance of coatings, the cross-sectional SEM morphologies and EDS line scanning of the uncoated N36 substrate and the AlNbTiZr MEA coatings with various Al content were implemented to estimate the depth of the oxide scale, as shown in Figure 9. All the coatings were well bonded to the substrates with a flat and compact morphology with blanket morphologies. Furthermore, there were no spallation or cracks between the corrosion regions and the lower coatings, which ascribed to (a) the low residual stress and (b) the high adhesive strength of the scale to the lower coating. The thicknesses of corroded coatings increased due to the formation of oxidation products, which was in agreement with the results obtained by Chen et al. after the corrosion of CrCuFeMoNi HEA coatings with different compositions [25]. Moreover, Figure 9h–l shows the cross-sectional EDS line scanning results of the coatings after corrosion. It can be seen that the oxygen signal was only detected on the external surface of the coatings and did not reach the substrate, which showed that no oxide was formed in the matrix, and all the coatings act as the barrier to oxygen penetration. This showed that the AlNbTiZr MEA coatings with different Al content have excellent corrosion protection properties in high-temperature pressurized water conditions. This is mainly due to the lack of a rapid diffusion channel for oxygen in the disordered amorphous structure of the coatings and the sluggish diffusion effect of MEA coatings, which inhibited the diffusion and migration of elements. As shown in Figure 10, the oxide scale depth of the corroded AlNbTiZr MEA coatings were compared with the uncoated substrate. The thickness of the oxide scale on the uncoated N36 substrate was the largest at 3.36 μm. The oxidation thickness of the Al10% coating was the lowest, which was only 0.84 μm. With the increase of Al content, the oxide scale thickness of the coatings first decreased and then increased slightly and was still much thinner than the oxide scale thickness of the uncoated N36 substrate. Meanwhile, the oxide scale depth of the Al-containing coatings was thinner than that of the Al-free coating, indicating that the addition of the Al element into the NbTiZr coating system can effectively improve the corrosion resistance. Compared with other Al-containing AlNbTiZr MEA coatings, the Al10% coating showed the best corrosion resistance, which was attributed to the composition ratio of lower Al content and higher Nb, Ti and Zr content in the Al10% coating. This can reduce the formation of the boehmite phase, and a denser oxide layer containing Nb_2_Zr_6_O_17_ and ZrO_2_ oxide products can be formed on the coating surface. Based on the above discussions, this method of reducing Al content can provide an idea for the design of a more optimal Al-containing MEA coating for ATF cladding.

## 4. Conclusions

We used magnetron co-sputtering on the AlNbTiZr MEA coatings with various Al contents on the N36 substrate. The effect of Al content on the microstructure and properties of the MEA coatings were investigated by SEM, XRD, TEM, nano-indenter and contact angle tester. The main conclusions are summarized as follows:The XRD and TEM analyses showed that all the coatings were composed of amorphous phase and tiny bcc nanocrystals; the number of nanocrystals in the coatings tends to increase with increasing Al content.With the increase of Al content in the coatings, the hardness and modulus of elasticity decreased first and then increased, while the cracking resistance and plastic deformation resistance increased slightly.Compared to the uncoated N36 substrate, the AlNbTiZr MEA coatings exhibited excellent corrosion resistance with a much thinner oxide layer and enhanced hydrophobicity.Compared with other coatings, the best corrosion resistance of AlNbTiZr MEA coating was obtained by the addition of 10.2% Al; because of its more reasonable composition ratio, it can greatly reduce the formation probability of the boehmite phase.

## Figures and Tables

**Figure 1 materials-14-07661-f001:**
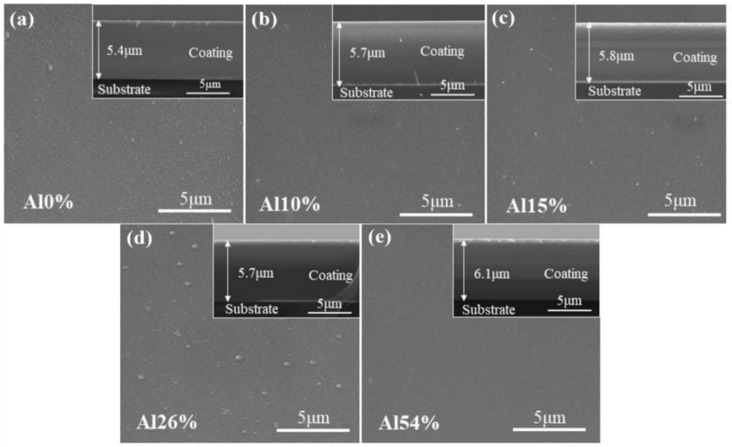
The surface and cross-sectional morphologies of the MEA coatings with various Al content: (**a**) Al0%, (**b**) Al10%, (**c**) Al15%, (**d**) Al26%, (**e**) Al54%.

**Figure 2 materials-14-07661-f002:**
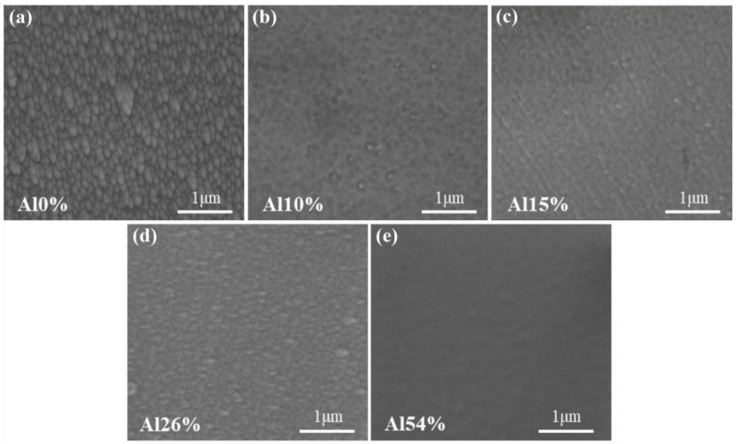
Surface SEM micrograph at high magnification of (**a**) Al0%, (**b**) Al10%, (**c**) Al15%, (**d**) Al26% and (**e**) Al54% coating.

**Figure 3 materials-14-07661-f003:**
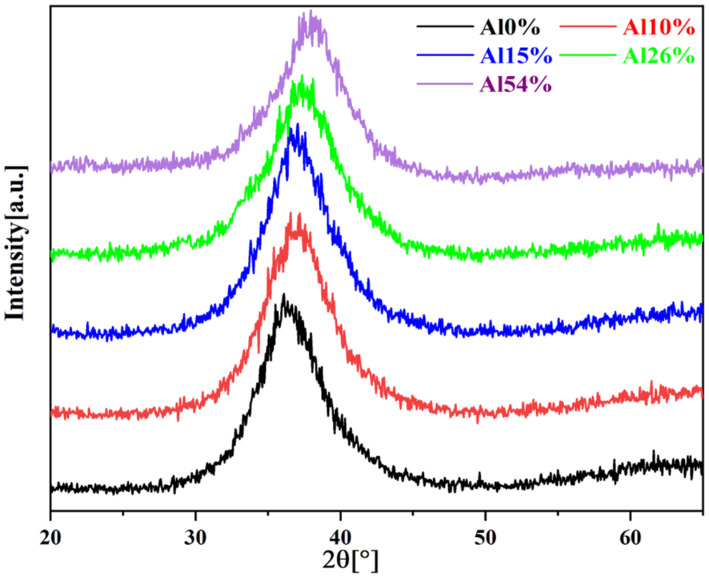
XRD patterns of the as-deposited AlNbTiZr MEA coatings with different Al contents.

**Figure 4 materials-14-07661-f004:**
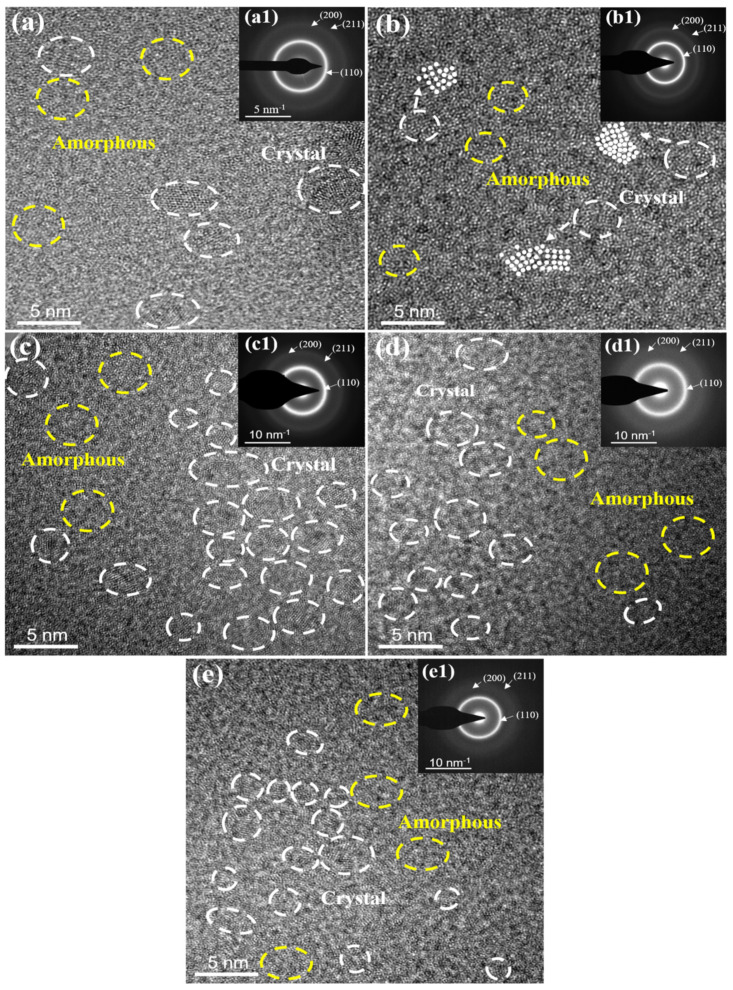
(**a**–**e**) The cross-sectional HRTEM images and (**a1**–**e1**) the corresponding SAED patterns of AlNbTiZr MEA coatings with increasing Al content.

**Figure 5 materials-14-07661-f005:**
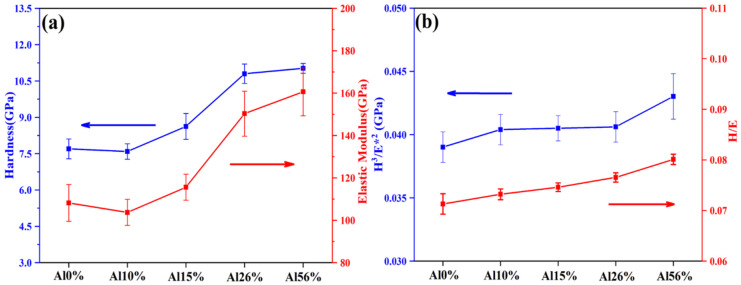
Hardness and elastic modulus (**a**) *H/E* and *H*^3^*/E**^2^ (**b**) of the Al0%, Al10%, Al15%, Al26% and Al54% coating.

**Figure 6 materials-14-07661-f006:**
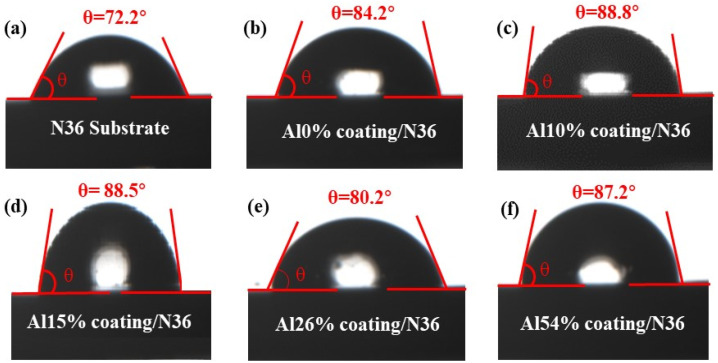
The apparent contact angles: (**a**) N36 substrate; (**b**–**f**) AlNbTiZr MEA coatings with different Al content.

**Figure 7 materials-14-07661-f007:**
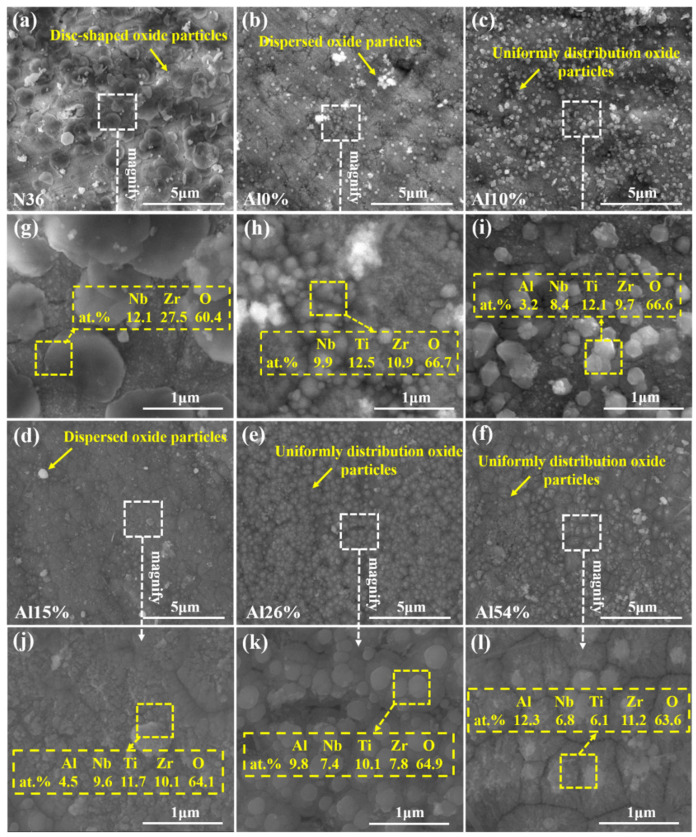
Surface SEM images of the N36 substrate and AlNbTiZr MEA coatings with different Al content after the autoclave tests: (**a**,**g**) N36 substrate; (**b**,**h**) Al0%; (**c**,**i**) Al10%; (**d**,**j**) Al15%; (**e**,**k**) Al26%; (**f**,**l**) Al54%.

**Figure 8 materials-14-07661-f008:**
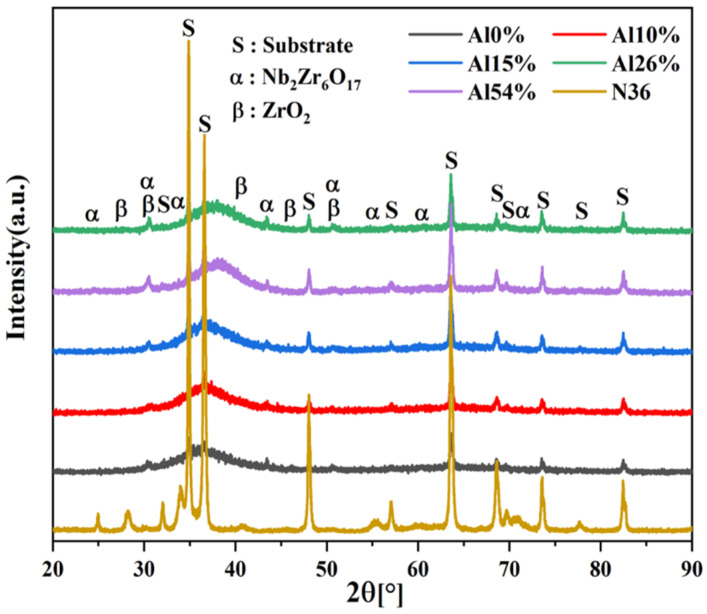
XRD patterns of the N36 substrate and the AlNbTiZr MEA coatings with different Al content after the autoclave test.

**Figure 9 materials-14-07661-f009:**
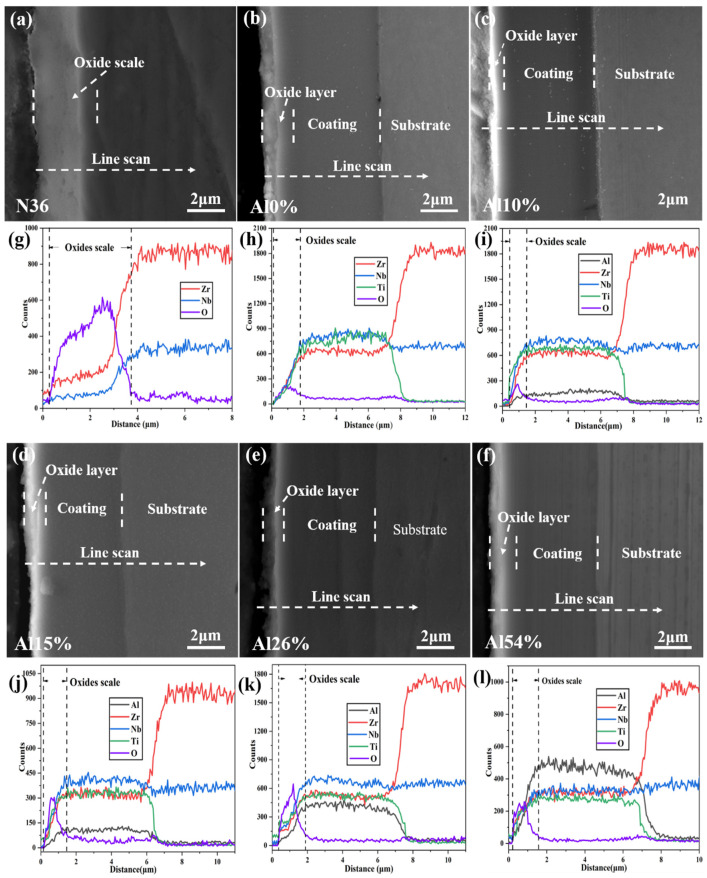
SEM cross-sectional images and corresponding EDS line scanning of the N36 substrate and AlNbTiZr MEA coatings with various Al content after the autoclave test: (**a**,**g**) N36 substrate, (**b**,**h**) Al 0%, (**c**,**i**) Al 10%, (**d**,**j**) Al 15%, (**e**,**k**) Al 26%, (**f**,**l**) Al 54%.

**Figure 10 materials-14-07661-f010:**
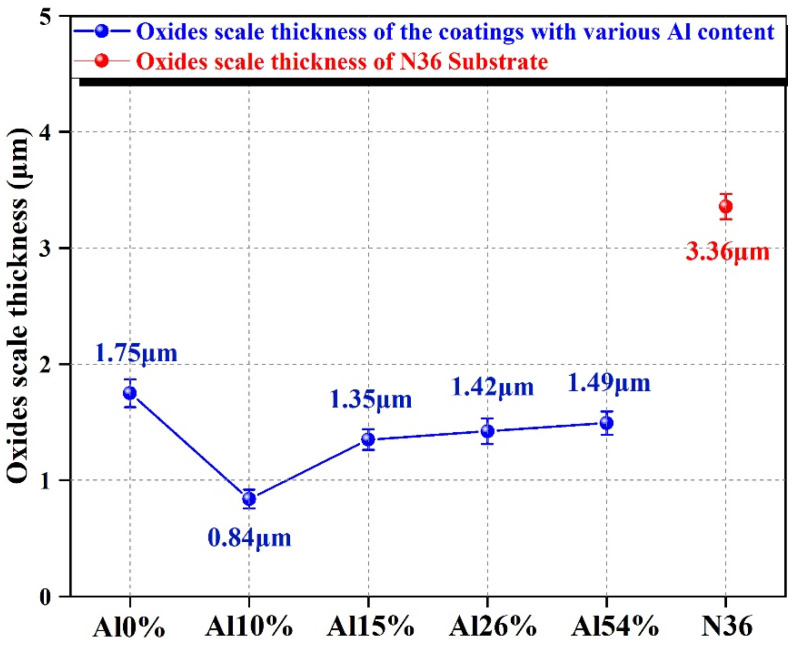
The oxide scale thickness of the AlNbTiZr MEA coatings with different Al content and the uncoated N36 substrate after autoclave test.

**Table 1 materials-14-07661-t001:** Experimental deposition parameters of the AlNbTiZr MEA coatings.

Base Pressure (Pa)	Working Pressure (Pa)	Working Temperature (°C)	Ar Flow Rate (sccm)	Substrate Bias Voltage(V)	Sputtering Power (W)	Deposition Time (min)
7.5 × 10^−4^	0.51	300	51	−100	200	150

**Table 2 materials-14-07661-t002:** The chemical compositions, thickness and mixing entropy of five MEA coatings.

Designated Name	Chemical Composition (at.%)				Coating Thickness (μm)	Mixing Entropy (J∙K^−1^ mol^−1^)
	Al	Nb	Ti	Zr		
Al0%	0	32.4	33.5	34.1	5.4	1.10*R*
Al10%	10.2	29.7	29.4	30.7	5.7	1.32*R*
Al15%	15.6	28.1	28.9	27.4	5.8	1.36*R*
Al26%	26.1	24.4	24.1	25.4	5.7	1.39*R*
Al54%	54.4	15.4	15.3	14.9	6.1	1.19*R*

## Data Availability

The data and results involved in this study have been presented in detail in the paper.

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
