# Peer review of "Effect of Al Concentration on Microstructure and Properties of AlNbTiZr Medium-Entropy Alloy Coatings"

_materials, 2021, doi:10.3390/ma14247661_

Round 1

Reviewer 1 Report

Review of paper no. materials-1462534 titled Effect of Al content on the microstructure and properties of AlNbTiZr medium-entropy alloy coatings by RF magnetron co-sputtering for accident tolerant fuel cladding by H. Xin et al.

This is an interesting and well-written paper that studies the effect of Al concentration on microstructure, phase constitution, hardness, wettability, and corrosion resistance of AlNbTiZr coatings. The materials were deposited on Si wafers and Zr alloy substrates by radiofrequency (RF) magnetron sputtering. The paper is acceptable for publication subject to minor revision.

1.The title is too long. It should be shortened as follows: Effect of Al concentration on microstructure and properties of AlNbTiZr medium-entropy alloy coatings

2.The chemical composition of the N36 zirconium alloy substrate (in wt.%) should be specified.

3.The dimension of mixing entropy is J K-1 mol-1. Please correct it in Table 2.

4.The red font on a black background in Fig. 4 is hardly visible. Use a more contrast color: yellow or orange, for example.

5.Oxide scale thickness is a more appropriate term than depth. Please correct it in Fig. 10 (legend and y-axis title).

Author Response

Response to Reviewer 1 Comments

Thank you for your letter and the referee’s comments concerning our manuscript entitled “Effect of Al content on the microstructure and properties of AlNbTiZr medium-entropy alloy coatings by RF magnetron co-sputtering for accident tolerant fuel cladding”.

We have carefully read the reviewer's comments and made necessary on our manuscript. In addition, the comments and suggestions from the reviewers are responded carefully. All response places were marked with red colour, and revised places in the manuscript with blue colour. I would re-upload the manuscript modified according to the comments of reviewers. The changes are listed below.

We hope that the paper in its revised form will be acceptable.

Sincerely yours,

Hongyang Xin

Point 1: The title is too long. It should be shortened as follows: Effect of Al concentration on microstructure and properties of AlNbTiZr medium-entropy alloy coatings

Response 1: Thanks very much for your careful and professional suggestions. I quite agree with your suggestion. I will update it in the manuscript and upload it again. To keep the context consistent, I still use "content" in the title. Please understand.

The revised title reads as follows:  

(I’ve marked it in the manuscript.)   Page 1, line 1 - line 2:

“Effect of Al content on microstructure and properties of AlNbTiZr medium-entropy alloy coatings”

Point 2: The chemical composition of the N36 zirconium alloy substrate (in wt.%) should be specified.

Response 2: Thanks very much for your careful and professional suggestions. I quite agree with your suggestion. I will update it in the manuscript and upload it again.

Specific modifications are as follows:

(I've marked it in the manuscript.)  Page 2, line 84 - line 86:

The five AlNbTiZr MEA coatings with different Al content were deposited on single-crystalline silicon wafers and N36 Zr alloy (Zr-1Sn-1Nb-0.3Fe in wt.%) substrates by RF magnetron co-sputtering.

Point 3: The dimension of mixing entropy is J K-1 mol-1. Please correct it in Table 2.

Response 3: Thanks very much for your careful and professional suggestions. I quite agree with your suggestion. I will update it in the manuscript and upload it again.

 (I've marked it in the manuscript.)  Page 4, line 149.

Point 4: The red font on a black background in Fig. 4 is hardly visible. Use a more contrast colour: yellow or orange, for example.

Response 4: Thanks very much for your careful and professional suggestions. I quite agree with your suggestion. I will update it in the manuscript and upload it again.

(I've changed it in the manuscript.)  Page 6, line 169.

Point 5: Oxide scale thickness is a more appropriate term than depth. Please correct it in Fig. 10 (legend and y-axis title).

Response 5: Thanks very much for your careful and professional suggestions. I quite agree with your suggestion. I will update it in the manuscript and upload it again.

(I've changed it in the manuscript.) Page 12, line 315.

Reviewer 2 Report

The submitted manuscript entitled "Effect of Al content on the microstructure and properties of AlNbTiZr medium-entropy alloy coatings by RF magnetron co-sputtering for accident tolerant fuel cladding" is a well-structured and clearly written paper, which suits in the Journal Materials. I recommend the manuscript for publication in its present form.

Author Response

Response to Reviewer 2 Comments

Thank you for your letter and the referee’s comments concerning our manuscript entitled “Effect of Al content on the microstructure and properties of AlNbTiZr medium-entropy alloy coatings by RF magnetron co-sputtering for accident tolerant fuel cladding”.

We have carefully read the reviewer's comments and made necessary on our manuscript. In addition, the comments and suggestions from the reviewers are responded carefully. All response places were marked with red colour, and revised places in the manuscript with blue colour. I would re-upload the manuscript modified according to the comments of reviewers. Some problems in the article have been corrected.

We hope that the paper in its revised form will be acceptable.

Sincerely yours,

Hongyang Xin

Reviewer 3 Report

Paper: Effect of Al content on the microstructure and properties of AlNbTiZr medium-entropy alloy coatings by RF magnetron co-sputtering for accident tolerant fuel cladding present some interesting experimental results on metallic layers deposited through magnetron co-sputtering with RF. The paper is interesting however present some problems: 

  • L 16: re-phrase The contact angle ......
  • L18: re-phrase 
  • L35: ref 5 is placed badly in the sentence 
  • L40: combine ) [20][21] [20,21] - please made this for the entire article 
  • L55: references 28, 29 and 30 appear in text before 27 
  • L63: intentionally designed : do you have work done not intentionally ? or what do you mean ? 
  • L84: Experimental details is second heading line (2)
  • L92: please explain: the substrates were performed ..... 
  • L98: secm ? 
  • L110: microscopy ...maybe is microscope 
  • L113: provide the type of EDS detector 
  • L121: how was the penetration depth .....? determined 
  • English language corrections are necessary 
  • Combine figure 1 with 2 , the SEM images from figure 1 are not helpful for the paper 
  • L158: re-phrase: Fig. 3 exhibited ....
  • L232: how many determinations were made for contact angle tests? are this average results ? 

Author Response

Response to Reviewer 3 Comments

Thank you for your letter and the referee’s comments concerning our manuscript entitled “Effect of Al content on the microstructure and properties of AlNbTiZr medium-entropy alloy coatings by RF magnetron co-sputtering for accident tolerant fuel cladding”.

We have carefully read the reviewer's comments and made necessary on our manuscript. In addition, the comments and suggestions from the reviewers are responded carefully. All response places were marked with red colour, and revised places in the manuscript with blue colour. I would re-upload the manuscript modified according to the comments of reviewers. The changes are listed below.

We hope that the paper in its revised form will be acceptable.

Sincerely yours,

Hongyang Xin

Point 1: L16: re-phrase The contact angle ......

 Response 1: Thanks very much for your careful and professional suggestions. I quite agree with your suggestion. I will update it in the manuscript and upload it again.

Specific modifications are as follows:

(I've marked it in the manuscript.)  Page 1, line 15 - line 16:

The hydrophobicity of the coatings was enhanced compared with that of substrate.

Point 2: L18: re-phrase

Response 2: Thanks very much for your careful and professional suggestions. I quite agree with your suggestion. I will update it in the manuscript and upload it again.

Specific modifications are as follows:

(I've marked it in the manuscript.)  Page 1, line 16 - line 18:

The 10.2 at. % Al contained AlNbTiZr coating had the best corrosion resistance, the minimum oxygen penetration depth, which originated from the formation of a denser oxide layer consist of Nb2Zr6O7 and ZrO2.

Point 3: L35: ref 5 is placed badly in the sentence 

Response 3: Thanks very much for your careful and professional suggestions. I quite agree with your suggestion. I will update it in the manuscript and upload it again.

(I've changed it in the manuscript.)  Page 1, line 34.

Point 4: L40: combine) [20][21] [20,21] - please made this for the entire article 

Response 4: Thanks very much for your careful and professional suggestions. I quite agree with your suggestion. I will update it in the manuscript and upload it again.

(I've changed it in the manuscript.)  Page 1, line 39.

Point 5: L55: references 28, 29 and 30 appear in text before 27

Response 5: Thanks very much for your careful and professional suggestions. I quite agree with your suggestion. I will update it in the manuscript and upload it again.

(I've changed it in the manuscript.)  Page 2, line 48 and line 53.

Point 6: L63: intentionally designed: do you have work done not intentionally? or what do you mean? 

Response 6: Thanks very much for your careful and professional suggestions. I quite agree with your suggestion. I will update it in the manuscript and upload it again.

(I've changed it in the manuscript.)  Page 2, line 61.

In our work, the novel AlNbTiZr MEA coatings with various Al content were designed and investigated for the ATF cladding coating.

Point 7: L84: Experimental details is second heading line (2)

Response 7: Thanks very much for your careful and professional suggestions. I quite agree with your suggestion. I will update it in the manuscript and upload it again.

(I've changed it in the manuscript.) 

Point 8: L92: please explain: the substrates were performed .....

Response 8: Thanks very much for your careful and professional suggestions. I quite agree with your suggestion. I will update it in the manuscript and upload it again.

(I've changed it in the manuscript.)   Page 2, line 91.

the substrates were polished by using different grit SiC abrasive papers

Point 9: L98: secm? 

Response 9: Thanks very much for your careful and professional suggestions. I quite agree with your suggestion. I will update it in the manuscript and upload it again.

(I've changed it in the manuscript.)   Page 2, line 97.

Point 10: L110: microscopy ...maybe is microscope

Response 10: Thanks very much for your careful and professional suggestions. I quite agree with your suggestion. I will update it in the manuscript and upload it again.

(I've changed it in the manuscript.)   Page 3, line 110.

Point 11: L113: provide the type of EDS detector

Response 11: Thanks very much for your careful and professional suggestions. I quite agree with your suggestion. I will update it in the manuscript and upload it again.

(I've changed it in the manuscript.)   Page 3, line 111.

Point 12: L121: how was the penetration depth ......? determined English language corrections are necessary   Combine figure 1 with 2, the SEM images from figure 1 are not helpful for the paper 

Response 12: Thanks very much for your careful and professional suggestions. I quite agree with your suggestion. I will update it in the manuscript and upload it again.

(I've changed it in the manuscript.)   Page 3, line 120- line121.

The oxides scale thickness of the coatings internal was tested to evaluate the corrosion resistance of AlNbTiZr MEA coatings with different Al content.

Figure 1 is to observe whether there are obvious holes, cracks and other defects on the coating surface from a relatively macro scale, while Figure 2 is mainly to illustrate the gradual process of the coating surface from rough to smooth with the increase of Al content. In order to ensure the overall structure and coherence of the article, I think it is necessary to continue to retain figure 1 and figure 2.

Point 13: L158: re-phrase: Fig. 3 exhibited ....

Response 13: Thanks very much for your careful and professional suggestions. I quite agree with your suggestion. I will update it in the manuscript and upload it again.

(I've changed it in the manuscript.)   Page 3, line 120- line121.

Fig.3 exhibits the XRD patterns of the as-deposited AlNbTiZr MEA coatings with different Al content.

Point 14: L232: how many determinations were made for contact angle tests? are this average results? 

Response 14: Thanks very much for your careful and professional suggestions. I will update it in the manuscript and upload it again.

(I've changed it in the manuscript.)   Page 7, line 217- line 218.

In order to reduce the experimental error, each sample was tested for 5 times and the average was taken.

We hope that the paper in its revised form will be acceptable.

Sincerely yours,

Hongyang Xin

Round 2

Reviewer 3 Report

Agree with publication in current form